# Frequency-aware Interface Dynamics with Generative Adversarial Networks

## Abstract

We present a new method for reconstructing and refining complex surfaces based on physical simulations. Taking a roughly approximated simulation as input, our method infers corresponding spatial details while taking into account how they evolve over time. We consider this problem in terms of spatial and temporal frequencies, and leverage generative adversarial networks to learn the desired spatio-temporal signal for the surface dynamics. Furthermore, we investigate the possibility to train our network in an unsupervised manner, i.e. without predefined training pairs. We highlight the capabilities of our method with a set of synthetic wave function tests and complex 3D dynamics of elasto-plastic materials.

## 1 Introduction

Complex and chaotic physical phenomena such as liquids, gels and goo are still very challenging when it comes to representing them as detailed and realistically as possible. A variety of numerical methods have been proposed to simulate such materials, from purely Eulerian methods (Harlow & Welch, 1965; Stam, 1999), over particle based methods (Gingold & Monaghan, 1977; Ihmsen et al., 2014), to hybrids (Zhu & Bridson, 2005; Stomakhin et al., 2013). Such simulations have also been targeted with deep learning methods (Tompson et al., 2017; Mrowca et al., 2018; Li et al., 2019), but despite significant advances, they remain very time-consuming and highly challenging to solve.

One approach to speed up the necessary calculations and to allow for more control is to employ *super-sampling*. This can be seen as a form of post-processing where one simulates only a low-resolution simulation and uses an up-sampling technique to approximate the behavior of a high-resolution simulation. Neural networks are of special interest here because of their capability to efficiently approximate the strongly nonlinear behavior of physical simulations. Applying neural networks to space-time data sets of physical simulations has seen strongly growing interest in recent years (Ladicky et al., 2015; Kim et al., 2020), and is particularly interesting in this context to incorporate additional constraints, e.g., for temporal coherence (Xie et al., 2018), or for physical plausibility (Tompson et al., 2017; Kim et al., 2019).

An important aspect here is that methods based on simple distance losses, such as mean square errors, quickly reach their limits. The generated data tends to be smooth without the necessary small-scale features. Generative adversarial networks (GANs) have been proposed to overcome this issue (Goodfellow, 2016). They are characterized by the fact that, apart from a generative network, they also make use of a discriminator that classifies the results of the generator with respect to the ground-truth data. Via a joint training, the distribution of solutions of the generator is guided to approximate the ground-truth data distribution. As the quality of the results is primarily determined by the discriminator network, it remains an open problem to accurately evaluate the quality of the inferred results. In our work we propose to evaluate the problem in the Fourier space. In this way, we are able to evaluate the given methods reliably, and it allows us to design improved learning algorithms that more faithfully recover the small scale details of the reference data.

For the core of our method, we build on an existing GAN-based architecture that employs two discriminator networks, one for the spatial and one for the temporal behaviour (Xie et al., 2018). In terms of ground truth data, we focus on multi-phase (solid-fluid-air) interactions with a sharp fluid-air interface. Unlike single-phase flow whose details are visible and relevant solely due to transparency throughout the volume, the details of our data are in most cases only visible on the surface. Of course, the internal dynamics in the volume also play a role, but they are mostly hidden

from the viewer, only the effects on the surface are visible. Furthermore, we consider phenomena that build up and take place over the course of several frames. Thus, as we will outline below, we employ a recurrent approach that is conditioned on a previous output in order to produce the solution for a subsequent timestep.

In order to represent and process fine details, we treat such detail as high-frequency displacements of a low-frequency surface, and correspondingly formulate the problem in Fourier space. The transformation into Fourier space yields an isolated view of the individual frequencies, and thus allows for a much improved analysis of the results achieved by different methods. E.g., it robustly identifies the strong smoothing behavior of $L_2$ metrics, and can detect mode collapse problems of adversarial training runs. We also demonstrate how frequency information can be incorporated into the learning objective in order to improve results.

To summarize, the central contributions of our work are: (1) A method for frequency evaluation with a consideration of spatial properties, (2) A novel frequency aware loss formulation, (3) A simple, yet intuitive evaluation of different generative methods, (4) A time consistent spatio-temporal up-sampling of complex physical surfaces.

**Related Work**  Deep learning methods in conjunction with physical models were employed in variety of contexts, ranging from learning models for physical intuition (Battaglia et al., 2016; Sanchez-Gonzalez et al., 2018), over robotic control (Schenck & Fox, 2018; Hu et al., 2019) to engineering applications (Ling et al., 2016; Morton et al., 2018). In the following, we focus on fluid-like materials with continuous descriptions, which encompass a wide range of behavior and pose challenging tasks for learning methods (Mrowca et al., 2018; Li et al., 2019). For fluid flows in particular, a variety of learning methods were proposed (Tompson et al., 2017; Prantl et al., 2017; Um et al., 2018). A common approach to reduce the high computational cost of a simulation is to employ super-resolution techniques (Dong et al., 2016; Chu & Thuerey, 2017; Bai et al., 2019). In this context, our work targets the up-sampling for physics-based animations, for which we leverage the approach proposed by Xie et al. (2018). However, in contrast to this work, we target phenomena with clear interfaces, which motivates the frequency-based viewpoint of our work.

For sharp interfaces, Lagrangian models are a very popular discretization of continuum mechanical systems. E.g., smoothed particle hydrodynamics (SPH) (Gingold & Monaghan, 1977; Koschier et al., 2019) is a widely-used particle-based simulation method. While points and particles are likewise frequently used representations for physical deep learning (Li et al., 2019; Ummenhofer et al., 2019; Sanchez-Gonzalez et al., 2020), Eulerian, i.e., grid-based representations offer advantages in terms of efficient and robust kernel evaluations.

We employ generative adversarial networks (Goodfellow, 2016), as a powerful and established method for learning generative models. Here, "unconditional" GANs typically rely on a synthetic input vector from Gaussian noise to produce the desired output distribution, e.g., the DC-GAN approach (Radford et al., 2016). Conditional GANs (Mirza & Osindero, 2014) were introduced to provide the network with an input that allows the neural network to steer the generation of the output. Hence super-resolution tasks for natural images (Ledig et al., 2016), or image translation tasks (Isola et al., 2017) employ conditional GANs. The time dimension was also taken into account in natural imaging works, e.g., by Saito et al. in the form of a temporal generator (Saito et al., 2017), or via a stochastic sequence generator (Yu et al., 2017). Other works have included direct $L_2$ loss terms as temporal regularizers (Bhattacharjee & Das, 2017; Chen et al., 2017), which, however, typically strongly restricts the changes over time. Similar to flow advection, video networks also often use warping information to align data over time (Liu et al., 2017; de Bezenac et al., 2017). We will demonstrate that recurrent architectures similar to those used for video super-resolution (Sajjadi et al., 2018) are likewise very amenable for physical problems over time.

## 2 METHOD

The input for our method is a coarsely approximated source simulation, with the learning objective to infer the surface of a target simulation over space and time. This target is typically computed via a potentially very costly, finely resolved simulation run for the same physical setup. When it comes to the possibilities of simulation representations, there is a great variance. In our case we have chosen an implicit representation of the data, by a signed-distance field (SDF) denoted by $g : \mathbb{R}^3 \rightarrow \mathbb{R}$.

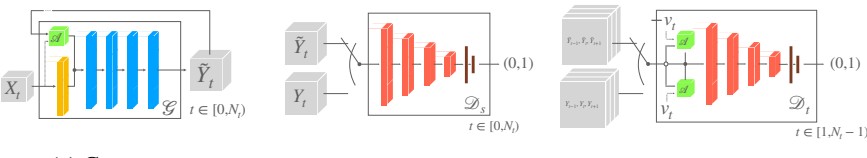

(a) Generator.  (b) Spatial & temporal discriminator.

Figure 1: The generator (a) takes low-resolution data $X_t$ as input, together with a high-resolution version, from the previous time step $\tilde{Y}_{t-1}$. The low-resolution data is tri-linearly up-scaled (orange layer), while the data with the previous state is advected with the velocity of the input (green layer). The processed inputs are concatenated and processed via four residual blocks (in blue). A high-resolution frame $\tilde{Y}_t$ is generated as output. (b) The structure of both discriminators (spatial and temporal) is similar: both use high-resolution data as input, either from the generator ($\tilde{Y}_t$) or ground-truth data ($Y_t$). The temporal discriminator additionally takes the previous and following frames and advects them with the velocity of the middle frame. Both discriminators process the data with four strided convolution layers (red layers) and two fully-connected layers (brown layers), combined with a spectral normalization. The output is the classification 0 (fake) to 1 (real).

An SDF returns, for a given point, the signed distance to the surface, with negative being inside the medium. Such a function is realized in practice by a grid $X \in \mathbb{R}^{M_x \times M_y \times M_z}$, storing the pre-computed signed distance values, where $M_*, * \in \{x, y, z\}$ specifies the size of the grid in the respective dimension $x$, $y$ or $z$. We have chosen this representation because most neural network layers are designed for array-like representations, and the loss functions on grid-based data are very efficient to evaluate. Additionally, an implicit representation via a grid can leverage tools from the field of level-set processing (Adalsteinsson & Sethian, 1999), and facilitates the frequency viewpoint via a Fourier transformation. Additional values, like the velocity, are also mapped on a grid $V \in \mathbb{R}^{M_x \times M_y \times M_z \times 3}$. Our goal is to let a generative network $\mathcal{G} : \mathbb{R}^{M_x \times M_y \times M_z \times 4} \rightarrow \mathbb{R}^{N_x \times N_y \times N_z}$ infer a grid $\tilde{Y}$ which approximates a desired high-resolution simulation $Y \in \mathbb{R}^{N_x \times N_y \times N_z}$ with $N_* = kM_*, N_* \in N_{\{x,y,z\}}$ and up-sample factor $k \in \mathbb{N}$, i.e. $\mathcal{G}(X) = \tilde{Y} \approx Y$. As our method only requires position and velocity data from a simulation, it is largely agnostic to the type of solver or physical model for generating the source and target particle data.

## 2.1 Neural Network Formulation

Our method is based on a generative, neural network with a 3D fully-convolutional ResNet architecture (He et al., 2016) that produces an output field at a single instance in time. The low-resolution input data is first up-sampled with a tri-linear up-sampling and then processed with several convolutional layers, as shown in Figure 1a. We use leaky ReLU as activation function after each layer, except for the last layer, where we use a $tanh$ activation. In our case, the input data consists of the implicitly represented geometry data $X_t$, the velocity $V_t$ of the simulation as well as the results of a previous pass $\tilde{Y}_t$. The previously generated data is advected with the low-resolution velocity before further processing. Through this feedback loop we train our network recurrently by iterating over a sequence of $T = 10$ frames. This yields stability over longer periods of time and gives better insights about temporal behaviour. Furthermore, the recurrent training is important to enable persistent behavior over time, such as the progression of fine surface waves. Unlike the process for generating the input data, the network training cannot resort to a physical simulation with full resolution, and hence cannot uniquely determine the evolution of future states. Therefore, its main learning objective is to capture the dynamics of the target simulations beyond that basic motion computed with an advection step. For initialization of the undefined first frame $\tilde{Y}_{-1}$ we use a tri-linear up-sampled version of the input. To train our network we have to define first a loss function that allows us to evaluate the differences between generated and ground-truth data. The most basic loss function is a simple mean squared error (MSE):

$$\mathcal{L}_s = ||Y - \tilde{Y}||_2^2. \tag{1}$$

This has the big disadvantage that it is ill-suited to measure the similarity or differences of solutions. For example, considering a function with multiple solutions for a given input, i.e., a multi-modal setting, a method that trains with an MSE loss will learn the expected value of the output distribution,

i.e, the average of the different solutions. However, the average is typically not a part of the solution set. Thus, the MSE loss often does not correspond to the correct distance in solution space, based on significant factors corresponding to the distribution of the solutions. Our super-sampling setup is such a problem: Due to the low resolution input, the high resolution details cannot be determined uniquely, resulting in a variety of possible solutions when up-sampling. Via physical properties of the material and its temporal sequence, some solutions can be eliminated, but nonetheless the space of solutions typically remains infinitely large. If an MSE loss is used, all such samples from the training data set are simply averaged to obtain a mean value, so that the result does no longer reflect the level of detail of the ground-truth data.

The MSE loss nevertheless gives a rough direction, and provides a stable learning target. Hence, we still use it as a component in the final loss formulation, in combination with an adversarial loss. In contrast to a direct distance metric, the adversarial loss approximates the ground-truth distribution. Hence, the network no longer learns one mean value, but chooses one valid solution out of the possible ones. We define a discriminator $\mathcal{D}_s$ that takes as input a high-resolution version of a simulation frame and classifies it, distinguishing between ground-truth and generated frames. It does this through a binary output, where 0 is "fake" and 1 is "real". Its task is to provide the generator with feedback on the correctness of the given data. The special feature is that the discriminators are trained together with the generator, thus creating a competitive interaction where both parties improve each other. As loss for the discriminator we use a binary cross-entropy:

$$L_{bce} = y \log(\tilde{y}) + (1 - y) \log(1 - \tilde{y}), \tag{2}$$

where $y$ is the ground-truth and $\tilde{y}$ is the generated value from the discriminator. For complex tasks, GANs can be unstable and difficult to control. For this reason we additionally use the recent Spectral Normalization (Miyato et al., 2018), which we found to provide more stable adversarial training.

While we have primarily focused on spatial content, i.e., the surface of the material so far, the temporal behavior likewise plays a crucial role, and poses similar difficulties in our multi-modal setting. On the one hand, the generation of details can quickly lead to temporally incoherent results, which is characterized by unappealing flickering. On the other hand, our network also should be able to match and recreate spatial solutions over time that reflect the physical behavior. Following previous work (Xie et al., 2018), we use an additional discriminator $\mathcal{D}_t$ to classify the temporal behavior of data. This is done by passing three corresponding frames, which are aligned with each other using advection $\mathcal{A} : \mathbb{R}^{N_x \times N_y \times N_z \times 3} \times \mathbb{R}^{M_x \times M_y \times M_z \times 3} \to \mathbb{R}^{N_x \times N_y \times N_z \times 3}$. Apart from this, the temporal discriminator closely follows the structure of the spatial discriminator. Both discriminators (Figure 1b) use a typical funnel structure, where the dimension is increasingly reduced using strided convolutional layers, with a last fully connected layer computing the classification result. We likewise use leaky ReLU activations, with a sigmoid function for the last layer.

The classification of the discriminators is included in the loss formulation of the generator:

$$\mathcal{L}_{Ds} = \frac{1}{T} \sum_{t}^{T} \mathcal{D}_s(\mathcal{G}(_tX), X_t),$$

$$\mathcal{L}_{Dt} = \mathcal{D}_t(\mathcal{A}(\mathcal{G}(X_{t-1}), V_t), \mathcal{G}(X), \mathcal{A}(\mathcal{G}(X_{t+1}), -V_t), X), \tag{3}$$

which gives the final loss function:

$$\mathcal{L}_G = \mathcal{L}_s + \alpha \mathcal{L}_{Ds} + \beta \mathcal{L}_{Dt}, \tag{4}$$

where $\alpha$ and $\beta$ indicate the weighting of the individual loss terms.

An additional benefit of the adversarial loss is that it allows for learning from unpaired data. A common problem for up-sampling methods is the generation of paired ground truth data for training. Due to different numerical approximations, and hence potentially differing physical behavior, the easiest solution is to simulate at high resolution, and down-sample the data. While at training time the down-sampled data is used, at test time, the model needs to be applied to data from a low-resolution simulation instead. This typically leads to large distribution shifts, and correspondingly impaired inference quality. Therefore, we take an unpaired training approach into account that decouples the low and high resolution data. The feedback from the discriminators is still based on the ground-truth data, which makes the output conditionally dependent on the input, but also approximates the behavior of the reference data. However, there is no direct supervision in the

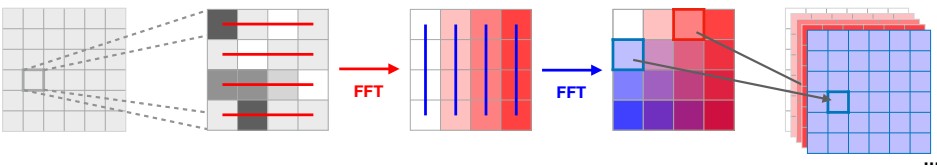

Figure 2: The process of our blockwise frequency evaluation. The input data is divided into blocks of a given size. A 2D/3D FFT is performed on the blocks, thus obtaining the individual Fourier coefficients, again in the form of grids. These grids are then arranged so that each grid corresponds to a Fourier coefficient.

generator anymore: the output is no longer compared with a matching ground-truth in the loss, but only related to the input. This is done by down-sampling the output and comparing it with the input:

$$\mathcal{L}_s^* = ||X - p(\tilde{Y})||_2^2, \tag{5}$$

where $p : \mathbb{R}^{N_x \times N_y \times N_z \times 3} \to \mathbb{R}^{M_x \times M_y \times M_z \times 3}$ is a down-sampling function based on average pooling. This effectively removes the need for paired low- and high-resolution samples at training time, and fully relies on the discriminator to match both distributions.

To indicate the focus on surface structures, we refer to the final version of our generative network as *surfGAN*. For a more detailed description of the training and the network architecture we refer to the appendix A.1 and A.2.

## 2.2 FREQUENCY EVALUATION

Given the formulation so far, an inherent difficulty that remains is a robust and reliable evaluation of the generated outputs. In a GAN setting, the discriminator determines the content, but it is typically not possible to evaluate whether it has correctly learned the shape of the output distribution. While obvious cases such as a mode collapse towards a constant signal are easy to detect, it remains challenging to reliably detect shifts in the data distributions, especially so for GANs (Arjovsky et al., 2017; Jolicoeur-Martineau, 2018). In our setting, the outputs should on the one hand represent a high-resolution version of the input, i.e., the down-scaled output should correspond to the input. This can be measured with a simple MSE. In addition, we also expect the generation of details that match the ground-truth as closely as possible. MSE is no longer usable for this, as even if the right details are generated, a slight translation would already lead to substantial errors. In addition, small-scale features have little effect on the MSE, despite being crucial for a realistic simulation result. Finally, the temporal behaviour should also correspond to that of the ground-truth.

Considering the problem in frequency space, the sought after detail consists of high-frequency features that cannot be represented by the low-resolution simulation. In addition, we have to distinguish between spatial and temporal frequencies. The frequency-based view has the advantage that it yields a simple but powerful performance evaluation that complements the discriminator of the GAN training. For the frequency evaluation we consider the SDF of our implicit grid $G_t \in \mathbb{R}^{N_x \times N_y \times N_z}$ for $t \in [0, T)$, which can be equated with the frequency behavior of the surface. In order to retain spatial properties in addition to the local frequency behaviour of the surface, we divide our image into blocks with size $b^3$ whose Fourier coefficients $g_{k_x, k_y, k_z}$ we evaluate, where $k_* \in [0, b)$. The method is similar to the one used for JPEG compressions. Therefore, we divide the domain into $N_x' \times N_y' \times N_z'$ regions for the spatial frequency evaluation, where $N_*' = N_*/b$. Using a three-dimensional fast Fourier transformation (FFT), we then transform the extracted blocks into the Fourier components:

$$g_{k_x, k_y, k_z} = \frac{1}{b^3} \left|\left| \sum_{n_x=0}^{b-1} \left( \omega^{k_x n_x} \sum_{n_y=0}^{b-1} \left( \omega^{k_y n_y} \sum_{n_z=0}^{b-1} \omega^{k_z n_z} \right) \right) \right|\right|, \tag{6}$$

where $\omega = e^{\frac{-i2\pi}{b}}$. An important point here is that we take the absolute value from the coefficients in order to eliminate the phase, and focus only on the amplitude of the frequency component. This

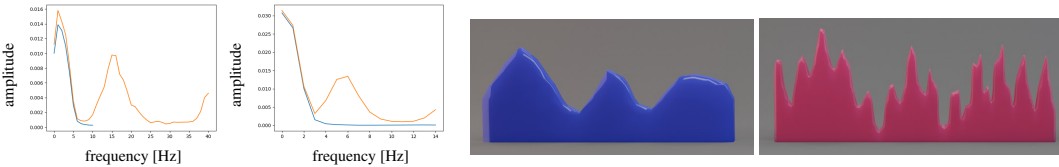

(a) Frequency histograms (spatial and temp.).

(b) Synthetic data sample.

Figure 3: The averaged frequency spectrum of our data set (a) for the surface, spatial (left), and for the temporal behavior (right). The spectrum of the low frequency input data is shown in blue, while in orange the spectrum of the ground-truth data is shown. In the spatial spectrum, it's clearly visible that the spectrum of the input data only covers a quarter of the frequencies, while a large peak of the ground-truth is visible for higher frequencies. This highlights what the generator needs to reproduce. A similar shape can be seen in the time spectrum as the time response is strongly coupled to the surface frequency. Here, the spectrum of the input data (blue) continues as the time discretization is the same for both. Figure (b) shows an example of a surface from our synthetic data set: the input wave in blue, and the corresponding ground-truth target in purple.

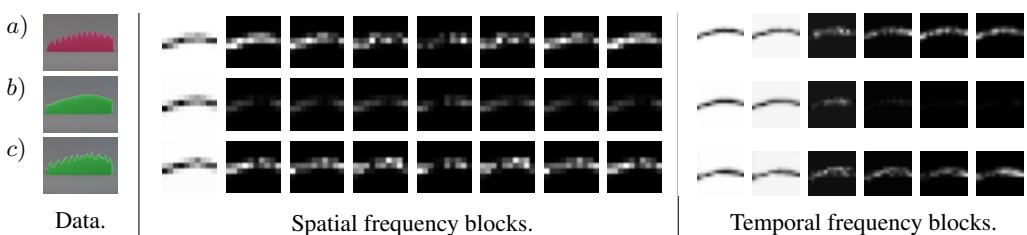

Figure 4: From left to right the image data and the corresponding spatial and temporal frequency images are shown. The respective columns indicate the frequency (from left, low to right, high frequency). We consider the ground-truth (a), a result based on MSE (b) and the surfGAN result (c).

makes the problem more robust for translated small-scale structures. Equation 6 yields $b \times b \times b$ Fourier components per box. The individual components can now be grouped with the same components from other blocks so that we get one $N_x/b \times N_y/b \times N_z/b$ grid per Fourier component (Figure 2), with each grid corresponding to a certain frequency. These grids can be further processed, e.g., via computational kernels, and in addition can be inspected by humans for verification. For the temporal frequencies we consider the changes per pixel over time via FFT. For longer sequences, a block-wise evaluation would be conceivable for time as well. The separation of spatial and temporal frequencies gives us the possibility to compensate for differences by adjusting the weighting. Based on this setup we can now evaluate results. On the one hand, we can directly compare the frequency range of generated data and identify missing details for single samples or mini-batches. E.g., this is amenable to loss formulations for learning objectives. On the other hand we can also create a histogram for the frequencies of the whole solution space and thus compare the frequency distribution of the generated and ground-truth data. This is especially helpful for GANs and works also for unsupervised setups.

## 3 RESULTS

For evaluation we consider two different data sets: a synthetic 2D case and a 3D particle-based simulation. The 2D data is fully controllable such that the frequency spectrum of the surface can be evaluated reliably. The basis is a wavy surface, based on a sine wave with varying frequency (Figure 3). This forms a wide range of analysis to isolate problems in the generative process and illustrate the aspects of the proposed method. We then evaluate the established methodology in 3D for a more complex scenario with data generated from a highly viscous SPH simulation (Weiler et al., 2018). For more information on both data sets, refer to the appendix A.3 and A.4.

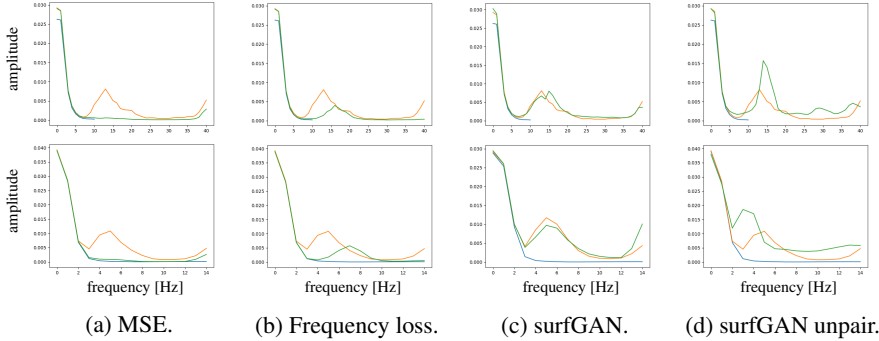

Figure 5: Frequency spectrum comparison of different versions of our network for the synthetic 2D data set. The top row always shows the spatial frequency, while the bottom row shows the temporal frequency. The spectrum of the input is shown in blue, the ground-truth in yellow, and the predictions in green. Versions (a) was trained with a simple MSE loss, while (b) was extended with a frequency loss; (c) and (d) were trained with an adversarial loss, whereby the latter evaluation was trained with unpaired data.

| | MSE | Frequency | surfGAN | surfGAN unpaired |
|---|---|---|---|---|
| **MSE** | 0.036 | 0.0361 | 0.0556 | 0.0389 |
| **Spatial freq. MAE** ($\times 100$) | 0.127 | **0.0711** | **0.0742** | 0.178 |
| **Temp. freq. MAE** ($\times 100$) | 0.302 | 0.128 | **0.0945** | 0.228 |

Table 1: Mean errors for 2D variants. The spatial and temporal MAE values represent the averaged difference of the respective frequency blocks.

## 3.1 FREQUENCY EVALUATION

We first consider the controlled 2D data. Training is performed with the full data set of 10000 samples with 30 frames, and we evaluate the resulting models on 10 simulations that are not part of the training set with 30 frames each. Examples of the block-wise analysis from our frequency evaluation (Figure 2) are shown in Figure 4. From top to bottom, it compares ground-truth, a network with MSE loss, and the surfGAN result. While the MSE-based variant is not able to reconstruct fine details, the surfGAN can recover these, as highlighted by the frequency evaluation. The high-frequency blocks clearly illustrate the differences into terms of reconstruction accuracy. In Table 1 and Figure 5 we also compare the mean error values of the different blocks quantitatively and via histograms. The mean errors likewise illustrate that the MSE version is not able to reconstruct the high frequencies of the target function. The spectrum of the generated data hardly shows any deflection in the high frequency range (Figure 5(a)). We additionally extended this fully supervised MSE loss with a term that takes into account differences in the frequency spectrum via an L2 norm between the Fourier components of the data. Figure 5(b) shows that this improves the situation, but does not suffice to reconstruct the full high frequency range. This is caused by the inherent averaging of the MSE, which is still suboptimal in Fourier space. In contrast, our surfGAN setup (Figure 5(c)) achieves the best results. The spectrum of the prediction very closely matches that of the ground truth. The GAN based method seems to be able to implicitly learn the underlying frequency distribution. Finally, we repeat the evaluation with an unpaired surfGAN setup. Despite the fundamentally more challenging learning setup, the network manages to recover the missing frequencies, as shown in Figure 5(d). Interestingly, the discriminator feedback causes the generator to slightly overshoot in terms of high frequency content.

| | MSE | tempoGAN | surfGAN |
|---|---|---|---|
| **MSE** | 0.0198 | 0.129 | 0.0404 |
| **Spatial freq. MAE** ($\times 100$) | 0.0259 | 0.039 | **0.0119** |
| **Temp. freq. MAE** | **0.206** | 0.477 | **0.205** |

Table 2: Mean error measurements for 3D test scenarios.

For our 3D results we have collected the values in Table 2. We compare our method with tempoGAN, whose performance suffers mainly from the missing recursion loop. It is clearly visible that surfGAN has the smallest error in the spatial frequencies, which correlates to the spatial detail level. The temporal frequency, on the other hand, is almost equal to the value of the NN with the MSE loss. This is probably due to the fact that the temporal behavior is relatively smooth.

## 3.2 QUALITATIVE RESULTS

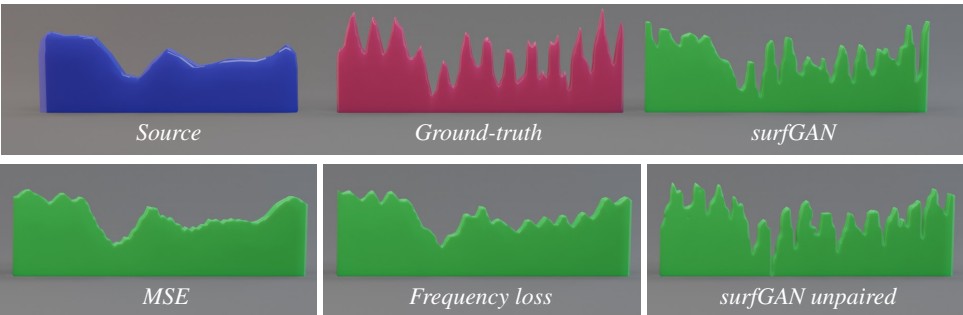

Figure 6: Comparison of a frame from a test run with the synthetic 2D data set. The input is shown in blue, ground-truth in red, and predictions in green.

As a qualitative evaluation we executed our network with test data and visualize the results. Figure 6 shows visual examples of the 2D frequency evaluation. While a version based on MSE generates very smooth images in the random data set, the surfGAN versions are able to reconstruct the jagged edges. In direct comparison with the ground-truth data there are differences, but this is due to the randomness of the data. Therefore, the exact solution cannot be reconstructed, but the surfGAN is still able to generate a very plausible solution.

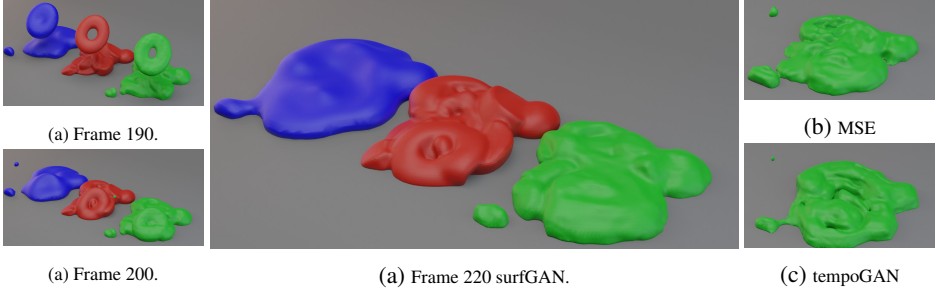

Figure 7: Comparison of 3D frames with a surfGAN model (a), with a MSE-based generator (b) and tempoGAN (c). The input is shown in blue, ground-truth in red, and predictions in green.

For the 3D case we focus on the paired surfGAN setup. Figure 7(a) shows that our prediction is able to reconstruct most of the details, even if they are not present in the input. We have deliberately chosen frames after a long run-time (190-220 time-steps) to show that details can persist and that they are the result of complex, physically-based behavior over multiple frames. Again, we compared our method with a simpler MSE-based generator and tempoGAN. With MSE, the results become comparatively smooth, as can be seen in Figure 7(b). In Figure 7(c) however, the results of tempoGAN are shown, where details are generated quite randomly and display a rather chaotic behavior over time. Due to the lack of a recurrent processing, the network cannot build an accurate internal state, which has the effect that the details are generated unnaturally.

## 4 CONCLUSION

We have presented a learning-based method to infer spatio-temporal detail to complex physical simulations. Our method puts special emphasis on high frequency content, and we present an approach

for assessing GAN-based outputs in terms of the generated spatial and temporal frequencies. Interestingly, our proposed surfGAN performs better than a direct supervision in terms in frequency space. Our method provides a first step towards evaluation and synthesis of physical space-time processes, and could be employed for other phenomena such as turbulence Ling et al. (2016) or weather Zaytar & El Amrani (2016). Furthermore, it will be interesting to employ it in conjunction with other frequency-based representations Sitzmann et al. (2020).

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

## A  APPENDIX

### A.1  IMPLEMENTATION DETAILS

The main part of the generator consists of four blocks containing two convolution layers and a residual connection. The feature count of the convolution layers per block is as follows: $[[8, 32], [64, 64], [32, 8], [1, 1]]$. The kernel size is $5 \times 5 \times 5$ for all layers. For the discriminators we use only convolution layers and no residual blocks. Both discriminators, spatial and temporal, have the same number of features per layer: $[8, 16, 32, 32]$. The fully-connected layer at the end of the layer consists of 64 and 1 neurons. The kernel size is $4 \times 4 \times 4$ for all layers. It is important to mention that for the first convolution of every network we do not use zero padding as usual but mirror padding. This is because we work with tiles from the training data and not with the complete frame and zero padding falsifies the values at the transitions between the tiles.

### A.2  TRAINING DETAILS

For the training we have implemented our network with the Tensorflow Framework. We use an Adam optimizer with a learning rate of 0.00001 and a batch size of 16 for the 2D tests and 4 for the 3D tests for 50k iterations. All other weights are initialized with the respective standard initializers of Tensorflow version 2.1. The weighting factors $\alpha$ and $\beta$ of Equation 4 are set to 1.0 and 10.0 correspondingly.

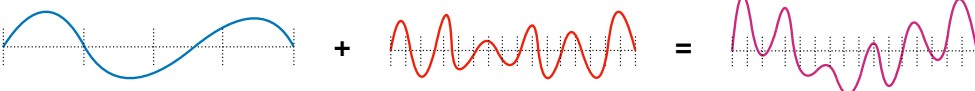

Figure 8: In blue a wave with a randomly varying low frequency $f_l$, which can still be represented by the low-resolution sampling and serves as input data set for our synthetic data set. In red, on the other hand, we see a wave with a high frequency $f_h$ that can only be correctly represented with much higher resolution. Combined with the low-frequency version, we get the ground-truth data (violet) for the synthetic data set.

### A.3  SYNTHETIC DATA SET

We use a synthetic data set to test and evaluate different aspects of our approach. The data set is designed to have a simple, clearly defined behavior, and such that the frequency spectrum of the surface can be evaluated reliably. Therefore, we use a horizontal wavy surface based on a 1D sine function $s_t(x)$ with randomly varying frequency $f_l \in (0, \frac{M}{2})$:

$$s_0(x) = sin(\frac{\pi f_l(x)x}{M}), x \in [0, M), \tag{7}$$

where $M$ is the resolution of the low-resolution data.

To make a time sequence out of this, we use a simple wave equation:

$$\frac{\delta^2 s}{\delta t^2} = \frac{\delta^2 s}{\delta x^2}, \tag{8}$$

Discretizing this we can calculate the vertical velocity of our surface as follows:

$$v_t(x) = v_{t-1}(x) + \frac{2 * s_t(x) - s_t(x - \Delta x) + s_t(x + \Delta x)}{\Delta x}, \tag{9}$$

where $v_0(x) = 0$. Given the velocity we can now calculate the next frame as follows:

$$s_t(x) = s_{t-\Delta t}(x) + \Delta t v_t(x), \tag{10}$$

For the high resolution data set, i.e, the targets to be learned, we use the same low resolution wave as base, and modulate it with a high frequency component (Figure 8):

$$t_0(x) = sin(\frac{\pi f_l(x)x}{kM}) + sin(\frac{\pi f_h(x)x}{kM}), x \in [0, kM), \tag{11}$$

where $k$ is the chosen up-sampling factor and the frequency $f_h(x)$ is chosen so that it cannot be represented by the low-resolution version. According to the Nyquist-Shannon sampling theorem, the frequency should be higher than $\frac{M}{2}$ and below $\frac{kM}{2}$ (Figure 3(a)). The generation of a sequence is done in the same way like for the low-resolution data.

Based on this setup, we generate two different data sets of high resolution data: one where we modulate with a fixed high frequency $f_h(x) = const$, whereas in the second version we vary the this high frequency component (Figure 3(b)). Thus the first version represents a deterministic up-sampling, which a generator should be able to reconstruct perfectly, whereas the second version is ill-posed, i.e. several solutions are possible, and hence poses a much more difficult learning target. Both data sets consist of 10000 sequences with 30 frames each at the end. While the deterministic version only serves as a sanity check which we only discuss here in the appendix (A.5) , the second, randomized version shows how well the method can approximate the ground-truth distribution for ill-posed tasks like the actual super-resolution problem for physical simulation data.

## A.4  SIMULATION DATA

For the generation of simulation, data we use an SPH solver of the SPlisHSPlasH framework (Bender, 2017). There are different materials to choose from. With materials that exhibit high-frequency physical behavior, chaotic behavior occurs in some cases, such as splashes in water, which are typically very difficult to reconstruct. With more viscous materials, such as gel, details are mainly distinguished by folds and fine waves on the surface. The chaotic behaviour is very difficult to reconstruct and it is sometimes very difficult to understand how correct the behaviour is. For these reasons we focus more on materials like gel. With gel, fine wrinkles can form on the surface which allows a good evaluation of the method. Another special feature is that details are persistent over time. Methods such as Xie et al. (2018) cannot represent such details because they do not provide feedback in the network. This means there is no memory. Therefore, we use an plastic material with high viscosity, simulated with an advanced viscosity solver (Weiler et al., 2018).

For the training we generate 60 different scenes with 300 frames per simulation. The time step corresponds to 100 frames per second. The scenes consist of randomly generated shapes that fall into a pool from different heights at random times. This creates interesting waves and folds on the surface. The ground-truth resolution is $160^3$. For the generation of the low-resolution data we distinguish between the way the training is done. For the supervised setup the ground-truth data is scaled down by the desired up-sampling factor $k$ and then smoothed with a Gaussian blur. This results in synchronous data pairs that can be used in a supervised setup. In the unsupervised setup, however, the low-resolution data is generated in the same way as the high-resolution data, with the correspondingly lower resolution. From the position and velocity data of the particles, we then generate our SDF and velocity grid which we use for training. Before the training, the data is normalized over the whole data set, so that the data is in the value range between -1 and 1. For a sharper edge in the SDF grid, which prevents unwanted noise in the generated data, we additionally modify the normalized values with a tangent hyperbolic function:

$$f(x) = \tanh(cx) \tag{12}$$

where $c$ can be freely selected, depending on the strength of the transition. We found 5 to be a good value. Finally, we take advantage of the locality of our problem and only use excerpts from the training data frames in training. On the one hand, this saves memory, because the data is sometimes very large and can cause problems with the GPU memory, on the other hand it allows us to extract only relevant parts of the data. So in our example we can only consider the data in places where there is a surface. Finally we have the advantage to augment the data by overlapping the tiles we extract, so we can get a lot of information from only a few frames.

## A.5  ADDITIONAL RESULTS

As a sanity check we used a deterministic setup as described in A.3. With this setup we tested if the network is able to modulate a static high frequency with a simple MSE loss. In Figure 9a this is shown to be the case. This serves as a baseline for what the network can achieve. In Table 9b we compare the frequency deviation with the MSE network trained with the non-deterministic setup. As expected, the error is much smaller with the deterministic setup.

In Figure 10 and Figure 11 we compare the frequency evaluation of two more data samples, the same as in Figure 4.

Finally there are some more 3D results in Figure 12 and Figure 13.

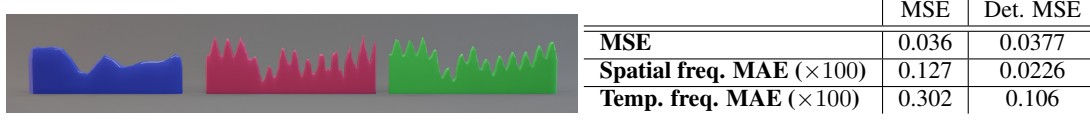

|  | MSE | Det. MSE |
|---|---|---|
| **MSE** | 0.036 | 0.0377 |
| **Spatial freq. MAE** ($\times 100$) | 0.127 | 0.0226 |
| **Temp. freq. MAE** ($\times 100$) | 0.302 | 0.106 |

(a) MSE with deterministic setup.    (b) Comparison det. vs not det.

Figure 9: Evaluation of deterministic setup.

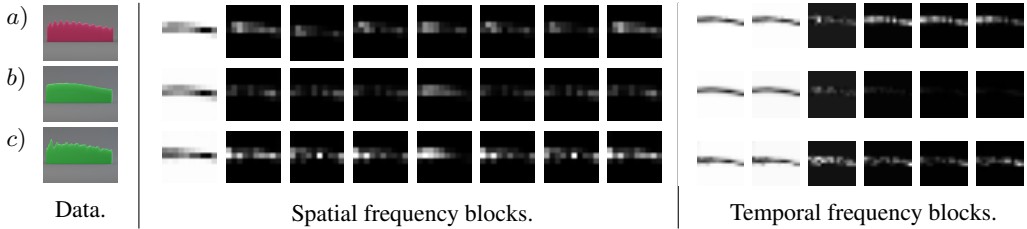

Figure 10: F.l.t.r the image data and the corresponding spatial and temporal frequency images are shown. We consider the ground-truth (1), a result based on MSE (2) and the surfGAN result (3).

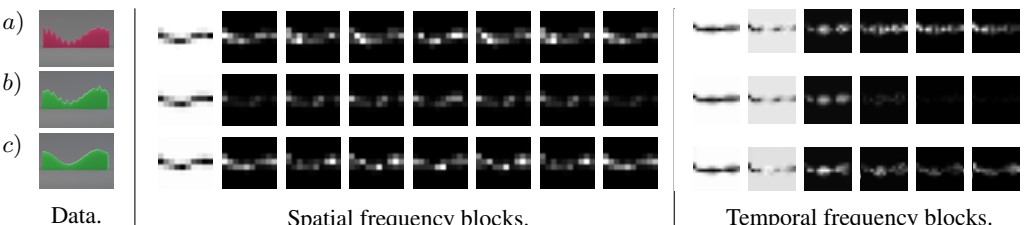

Figure 11: F.l.t.r the image data and the corresponding spatial and temporal frequency images are shown. We consider the ground-truth (a), a result based on MSE (b) and the surfGAN result (c).

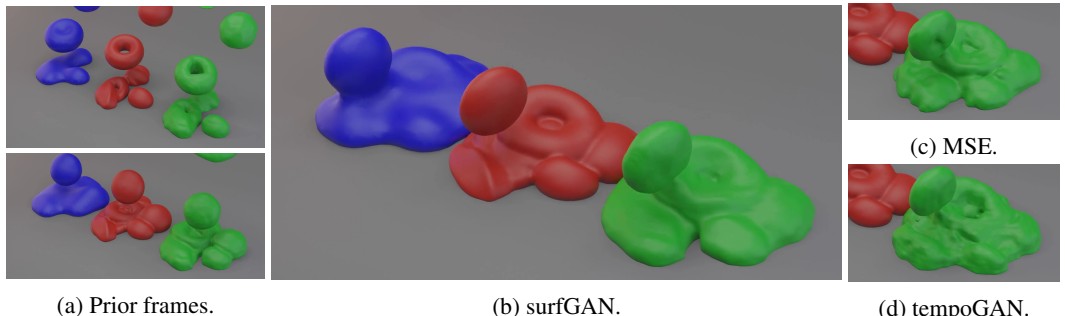

(a) Prior frames.      (b) surfGAN.      (c) MSE.

(d) tempoGAN.

Figure 12: Comparison of a 3D test run wit our surfGAN (b), with a MSE-based generator (c) and tempoGAN (d). The input is shown in blue, ground-truth in red, and predictions in green.

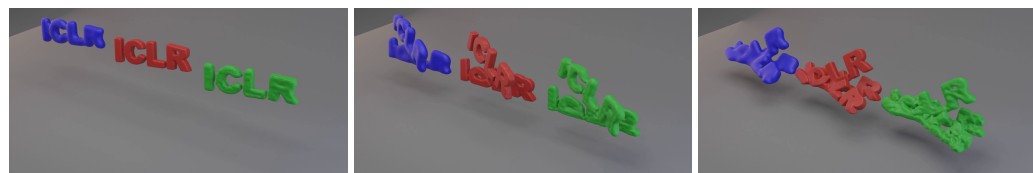

Figure 13: An additional output generated with the surfGAN network. The input is shown in blue, ground-truth in red, and the prediction in green.

