# OpenReview forum: "Frequency-aware Interface Dynamics with Generative Adversarial Networks"
_ICLR.cc/2021/Conference — Reject_

### Official Review · AnonReviewer3 · 2020-10-27
**interesting approach using GAN for physical simulation, while method lacks novelty and results are quite preliminary**

**Rating:** 4
**Confidence:** 3

**Review:**

**Summary**
This paper presents a GAN framework to learn spatial and temporal representation on complex physical surfaces to apply to simulation. The method represents data in a SDF-like way so it is agnostic to the properties of material and simulation model. The network is built upon conventional GAN, with two discriminator for temporal and spatial. A loss function is proposed to evaluates on a grid-based Fourier transform of the output and ground truth to better preserve high frequency details (temporally and spatially). Results show performance on a physical simulation with opaque, elasto-plastic materials.

**Strength**
Using generative model for physical simulation is an interesting task. Physical simulation contains high temporal frequency / variance that would benefit from GAN that is known to preserve high frequency features.

**Weakness**

    - Method on Fourier domain supervision lacks more analysis and intuition. It's unclear how the size of the grid is defined to perform FFT, from my understanding, the size is critical as local frequency will be changed using different grid size. Is it fixed throughout training? What is the effect of having different sizes?
    - The generator has a recurrent structure that supports 10 frame generation, but the discriminator looks at three frames (from figure 1) at a time, which seems to limit the power of temporal consistency.
    - In figure 7 result and supplemental video result, SurfGAN produces smoother results (MSE seems closer to the red ground truth in figure 7). This seems contradicts the use of Fourier components for supervision -- what causes this discrepancy?
    - Figure 4 is confusing. It's not clear what the columns mean -- it is not explained in the text or caption.
    - Notation is confusing. M and N are used without definition.

**Suggestion**

    - Spell out F.L.T.R in figure 4
    - Figure 1 text is too small to see
    - It is recommended to have notation and figure cross-referenced (e.g. M and N are not shown in the figure)

---

> ### Author Response · Authors · 2020-11-24
> **Response to Reviewer 3**
>
> *“Method on Fourier domain supervision lacks more analysis and intuition.”*
>
> This is a very good question, and we will definitely discuss it in more detail in our paper. Since the focus is mainly on frequencies that the low-resolution version cannot display, we have chosen the block size accordingly. Of course, you could also choose larger blocks, but then the locality will be lost and the calculated values can be less well assigned to certain spatial areas. I.e. the final frequency image would be smoother and less detailed.
>
> *“... discriminator looks at three frames ...”*
>
> We had tested a different number of input frames and had the best results with three frames. Basically, it is part of the hyperparameters, where the number of recursions in recursive training is more useful for generating long-term effects, while the input to the temporal discriminator is more for temporal coherence, where three frames seem to be enough. We will consider integrating a corresponding evaluation in a future version.
>
> *“MSE seems closer to the red ground truth in figure 7”*
>
> In fact, the differences are not so easy to see in the picture, because it is a momentary shot. But in the video in the appendix, you should see a very strong temporal incoherence and also a physically unrealistic behavior for the results with MSE. We understand that a visual evaluation is sometimes also relatively subjective, so we hoped to get a more objective view by the frequency evaluation which we provided.
>
> *“Figure 4 is confusing. It's not clear what the columns mean -- it is not explained in the text or caption.”*
>
> This is a very good point, we will add the appropriate labels. Along the x-axis, we show different frequencies (from low to high frequency).
>
> *“Notation is confusing. M and N are used without definition.”*
>
> Thank you for pointing this out, we will add to it.

---

### Official Review · AnonReviewer4 · 2020-10-28
**The authors use adversarial losses over space and time for simulation refinement. Incremental novelty in approach.**

**Rating:** 3
**Confidence:** 3

**Review:**

The authors propose to use the same formulation of adversarial losses over space and time as done by Xie et. al. The block-wise frequency evaluation is a new contribution of the paper.

The paper reads more like a report as opposed to showing unique insights for the niche problem the authors have tackled. There is little to no novelty in the MSE and adversarial loss formulation over space and time. The use of a generator and two discriminators is exactly the same as done in tempogan. I can see two differences:
1. Use of MSE as additional loss
2. Use of generator output from previous time step as a recursive input.

Both these additions are incremental and not novel. The evaluation is contrived in my opinion. It is natural that tempogan will underperform wrt to MSE metric. The frequency based metrics are derived by the authors and the sparse qualitative comparison to tempogan on a single example make it hard to be confident about the improvement over tempogan. It needs to be evaluated both qualitatively and quantitatively on a wider range of applications like done in tempoGan to validate its usefulness. I would like evalaution with more neutral metric which exists in literature beyond the ones in the paper. As the evaluation metric is derived by the authors, it is hard to validate the benefit of the approach and the evaluation protocol.

There should be ablation studies that discuss the suggested improvements in architecture over tempogan. Also the authors should compare with simple baselines like nearest neighbor and encoder-decoder networks. As the MSE loss is evaluated over paired samples,  the usefullness of GAN should be validated in the paired case. Some discussion on directly optimzing the frequency metric proposed in the paper using a GAN will be useful to the readers. Overall, the originality of the work is very weak and experiments are insufficient.

Post rebuttal comments: Thank you for the rebuttal. The broad concerns of insufficient novelty remain and I am sticking to my initial paper rating.

---

> ### Author Response · Authors · 2020-11-24
> **Response to Reviewer 4**
>
> *“The use of a generator and two discriminators is exactly the same as done in tempoGAN. I can see two differences…”*
>
> Apart from that, we tried to remove the limitation of tempoGAN for the intended task in the generative part. The main changes are, to make it clear once again:
> - a recursive network to generate temporally consistent details.
> - the possibility of unpaired training.
> - a frequency-based evaluation of our network.
> - the use of a variation of SN-GAN instead of the simpler BEGAN used in tempoGAN.
> - focus on surface-based data (two-phase flows), while tempoGAN only considers single-phase flows (based on density)
> MSE was also used by tempoGAN as an additional loss, in this, we do not differ.
> However, since the distinction to tempoGAN is not yet strong enough, we will try to differentiate ourselves even more from tempoGAN in a future version of the paper and show the differences through more detailed evaluations.
>
> *“ I would like evalaution with more neutral metric which exists in literature beyond the ones in the paper.”*
>
> We would like to make further evaluations with neutral metrics, but we are not quite sure which ones could come into question. PSNR is one possibility, but it should have similar weaknesses to an MSE loss. We would appreciate it if reviewers could provide an example of a metric that we have not considered so that we can expand our work accordingly. In our eyes, it is difficult to find a metric that meaningfully reflects the quality of the results, which is why we had defined our own metric.
>
>
> *“As the evaluation metric is derived by the authors, it is hard to validate the benefit of the approach and the evaluation protocol.”*
>
> It was important to us in our work to focus on surface frequencies, as we believe it is a good evaluation metric. In theory, a frequency-based loss could even take over the task of a discriminator, but it is still outperformed by the GAN as shown (Figure 6). Because of this, we use frequency-based loss mainly as an evaluation metric, which, as we have shown, provides a good insight into the performance of the GANs. The intention was to have a geometric counterpart of metrics like the Frechet Inception Distance. We found it very interesting to show the reconstruction capabilities of GANs in the frequency domain, which are surprisingly good if the hyperparameters are well adjusted. Cases, where there was a mode collapse or an imbalance between generator and discriminator, could be easily identified by the frequency evaluation.
>
>
> *“Some discussion on directly optimizing the frequency metric proposed in the paper using a GAN will be useful to the readers.”*
>
> This is a good point. In fact, we had done similar tests, but we didn't get better results, so we didn't see any added value in them. Nevertheless, we realize that for the sake of completeness it would be useful to add these results to the paper.

---

### Official Review · AnonReviewer2 · 2020-10-28
**Generation of complex surfaces with spatial and temporal discriminators under cycle-consistance supervision of down-sampling reconstruction loss**

**Rating:** 5
**Confidence:** 4

**Review:**

This paper introduces the adversarial generative network into a physical simulation task, i.e., reconstruction and refining of complex surfaces. Several losses are applied in the proposed framework, such as MSE, adversarial losses, low-resolution reconstruction loss. Experimental results seem good.

Pros:
+ As far as I know, this is the first work that uses GANs to refine complex surfaces.
+ Some specific changes have been made in the proposed framework. Specifically, MSE loss is introduced to help the low-frequency reconstruction, a spatial discriminator and also a temporal discriminator are used during the training progress, the adversarial losses also provide the ability of learning from unpaired samples, and to overcome the shortage of unpaired samples that cannot learn an exactly map function, a down-sampling reconstruction loss is applied as a cycle-consistence loss. All these loss functions look reasonable.
+ This paper is well-written and easy to read.

Cons:
- The main shortage of this paper is the experimental part. First, the comparison method is too weak, several good GAN frameworks, e.g. WGAN-GP, PU-GAN, and SN-GAN, are proposed and none of them is compared. Second, many loss functions are introduced in the proposed method. However, how these losses reflect the experimental results is not well studied. An ablation study is desired.
- The frequency part is interesting. But I am not sure if I miss something, I do not understand how the method leverage the frequency? Do you use the frequency block for reconstruction loss, or only use it to guide the training progress? I am confused about this.
- Although many loss functions are introduced, the proposed method is not novel. Most of the losses have been proposed previously. I think the contribution of this paper is not sufficient. However, I would like to read the response from the author about this, especially the frequency part.

---

> ### Author Response · Authors · 2020-11-24
> **Response to Reviewer 2**
>
> *“First, the comparison method is too weak, several good GAN frameworks, e.g. WGAN-GP, PU-GAN, and SN-GAN, are proposed and none of them is compared.”*
>
> We based our choice of the GAN setup mainly on the evaluation in the respective papers, where SN-GAN stood out as one of the best, which we then used. We had originally also tried the WP-GAN, but it did not perform as well, so we decided to use the SN-GAN until the end. The PU-GAN, on the other hand, requires a lagrangian-based representation, which does not support the planned focus on surfaces, so we had not considered it. We can understand that a comparison with other GANs would be useful, but we did not see this as the highest priority, because the GAN we used already gave good results. However, we will definitely think about a possible extension in the direction for the future.
>
>
> *“Second, many loss functions are introduced in the proposed method. However, how these losses reflect the experimental results is not well studied. An ablation study is desired.”*
>
> A good point. We had seen the evaluation with the 2D tests as a kind of ablation study, where we started with a simple MSE loss, which we extended with a frequency loss or the discriminative loss. Since we had already excluded the frequency loss in this synthetic test, we concentrated on comparing the 3D results with MSE. One missing point is an evaluation with the spatial discriminator alone, which we will add in the future.

---

### Author Response · Authors · 2020-11-24
**Update in the Revision.**

We thank the reviewers for their detailed and helpful feedback. We will answer the individual reviews in order to clarify as many ambiguities as possible.
For our revision we have changed the following things:
- better defined the variables M and N
- in Figure 4 an explanation for the columns is given
- enlarged the text in Figure 1

---

### Decision · Program_Chairs · 2021-01-07
**Final Decision**

**Decision:**

Reject

**Comment:**

This paper addresses the problem of super-resolution of coarse physical simulations into fine-grained video by satisfying some physical properties. The method uses generative models for sequences of images (conditional GANs with spatial and temporal discriminators), that take into account both the multiplicity of realisations given the same conditioning (via adversarial losses) and the spatial frequencies of the modeled surfaces using an evaluation in the Fourier domain, and that also allow training without paired low- and high-resolution samples by relying on a generative losses applied to the downsampled reconstruction. As demonstrated on synthetic 2D data or on 3D frames showing particle simulations, the proposed method can generate images with high-frequency content.

Reviewers have praised the intuition of using GANs for modeling high-frequency features in physical simulation reconstruction. Weaknesses included:
* insufficient analysis and explanation of the Fourier-domain supervision
* limited novelty w.r.t. TempoGAN (with the exception of frequency-based evaluation) and lack of evaluation on more general tasks
* unconvincing results with marginal improvement over plain MSE loss
* missing ablation studies
* toy dataset evaluation only, which makes this work seem preliminary

This paper comes out as 6/7 in my AC stack and I will unfortunately recommend to reject it, hoping that the authors will resubmit an improved version, taking into account the reviewers’ suggestions, for an upcoming conference or for a journal venue.